# Unlocking Transfer Learning for Open-World Few-Shot Recognition

**Byeonggeun Kim**[*†]**, Juntae Lee**[*1]**, Kyuhong Shim**[†]**, and Simyung Chang**[†]
[1]Qualcomm AI Research[‡]
`{juntlee}@qti.qualcomm.com`

## Abstract

Few-Shot Open-Set Recognition (FSOSR) targets a critical real-world challenge, aiming to categorize inputs into known categories, termed closed-set classes, while identifying open-set inputs that fall outside these classes. Although transfer learning where a model is tuned to a given few-shot task has become a prominent paradigm in closed-world, we observe that it fails to expand to open-world. To unlock this challenge, we propose a two-stage method which consists of open-set aware meta-learning with open-set free transfer learning. In the open-set aware meta-learning stage, a model is trained to establish a metric space that serves as a beneficial starting point for the subsequent stage. During the open-set free transfer learning stage, the model is further adapted to a specific target task through transfer learning. Additionally, we introduce a strategy to simulate open-set examples by modifying the training dataset or generating pseudo open-set examples. The proposed method achieves state-of-the-art performance on two widely recognized benchmarks, miniImageNet and tieredImageNet, with only a 1.5% increase in training effort. Our work demonstrates the effectiveness of transfer learning in FSOSR.

## 1 Introduction

Few-shot learning (FSL) has got a lot of attention due to the importance in enabling models to adapt to novel tasks using a few examples (e.g., $N$-way $K$-shot: a task involving $N$ distinct classes, each represented by $K$ examples) [42, 17, 37, 34, 27, 10]. However, in practical applications, FSL models inevitably encounter instances that do not belong to the $N$ classes, also known as open-set instances. Addressing this challenge has led to the emergence of the field of few-shot open-set recognition (FSOSR) [24, 16, 19, 15, 41].

In FSOSR, if two $N$-way $K$-shot FSOSR tasks have distinct closed sets, their corresponding open sets will also differ. This interdependence presents a key challenge when adapting to novel tasks in FSOSR. Namely, it is essential to redefine not only the closed set but also the open set, since the open set is inherently shaped by its closed set. Consequently, the open set lacks a universal definition across various FSOSR tasks; instead, it requires contextual consideration based on the closed set of a specific target task.

Despite recent advancements in the field, current works [24, 16, 19, 15, 41] have commonly focused on leveraging prior knowledge from a large training dataset. Then, they frequently struggle to balance closed-set accuracy with open-set recognition capabilities, often prioritizing open-set recognition at the expense of closed-set accuracy. Then, FSOSR research have faced saturated performance and struggled to achieve broad generalization across various benchmarks. In this work, we bring at-

---

* Equal contribution
†Work completed during employment at Qualcomm Technologies, Inc.
‡Qualcomm AI Research is an initiative of Qualcomm Technologies, Inc.

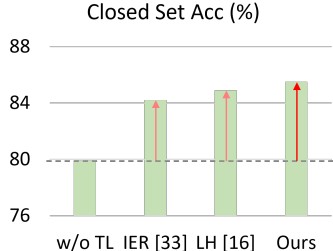 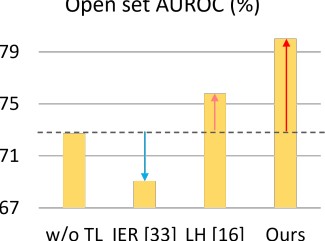

Figure 1: Difficulty of straightforward extension of the transfer learning from FSL methods [34, 17] to FSOSR. Compared to the pre-trained model without transfer learning (w/o TL), in open-set recognition, [34, 17] are less effective as much as in closed-set, or even degrade the performance.

tention to the novel application of transfer learning within this field. Transfer learning [14, 7, 6, 30] has been extensively studied and demonstrated its efficacy leveraging a pre-trained model to generalize it to other tasks. Recent FSL methods [31, 34, 39, 17, 35] have shown the efficacy of this approach. However, when they come to FSOSR, open-set examples are inherently not present, which significantly undermines the effect of transfer learning in terms of open-set recognition. Then, as in Fig. 1, the naive extension of the transfer learning techniques of FSL, e.g. IER-distill [34] and Label Halluc. [17] fails to attain the same level of improvement in open-set recognition as seen in closed set, or even results in decreased result.

Tackling this point, we propose a two-staged FSOSR learning framework. Our method involves two stages: *open-set aware meta-learning (OAL) and open-set free transfer learning (OFL)*. During the meta-learning stage, our objective extends beyond the meta-training of the feature encoder; we also aim to establish a universal open-set representation. This equips us with a decent starting point for the subsequent open-set free transfer learning. In the transfer learning stage, we commence by initializing the model using the parameters obtained from the meta-learning stage. To counteract the absence of open-set examples, we develop two alternative open-set sampling strategies. The first approach curates a training dataset of the previous stage as a source of open-set examples for open-set free transfer learning. For more pragmatic application, our second strategy is confined to the closed-set examples present in the target task, and exploits an episodic learning framework. Here, we generate *pseudo* FSOSR episodes by randomly dividing the closed-set categories into a closed set and a pseudo open set. As a result, our OAL-OFL method attains a marked enhancement in performance metrics on the standard FSOSR benchmarks as depicted in Fig. 1 while incurring a minimal extra training expense of only 1.5% compared to training without transfer learning. This allows OAL-OFL to surpass the existing state-of-the-art (SOTA) methods.

**Our contributions**: **i)** We introduce a novel two-staged learning called OAL-OFL, bringing transfer learning to FSOSR for the first time with only a minimal additional cost. **ii)** We show the importance of preparing the model through open-set aware meta-learning, which is a sturdy starting point for transfer learning. **iii)** We suggest two breakthroughs to handle the lack of open-set examples during the transfer learning stage. **iv)** By leveraging the effectiveness of transfer learning, our proposed OAL-OFL achieves SOTA on *mini*ImageNet and *tiered*ImageNet datasets. This underscores its ability to generalize across various tasks, enhancing both closed-set classification accuracy and open-set recognition capabilities.

## 2   Related Works

**FSL** [9] can be streamlined into two approaches: meta-learning and transfer learning. Meta-learning, also known as learning-to-learn, can be categorized into optimization and metric-based methods. Optimization-based methods [10, 11, 22, 27] involve training a meta-learner to adapt quickly to new tasks through a few optimization steps by learning adaptation procedures. In contrast, metric-based methods [20, 37, 40, 42] use a common feature embedding space where target classes of a new task can be distinguished using a distance metric. Recently, transfer learning-based approaches [31, 34, 17, 35] have been suggested with superior results where a model trained on a large-scale base dataset is fine-tuned to a few-shot task with a small number of support examples. Our approach aligns with the transfer learning-based approaches of the few-shot classification, but transfer learning has yet to be explored in the context of FSOSR.

**OSR** aims to identify in-distribution samples within a closed set, and simultaneously detect the out-of-distribution samples from unseen classes. Early approaches focused on the confidence level and proposed various approaches such as using extreme value theory [1, 2, 13, 36]. Recently, there has been a growing interest in generative model-based approaches [28, 29, 38], which leverage the different reconstruction behavior between in-distribution and out-of-distribution samples. On the other hand, our method aligns more closely with distance-based approaches [18, 25, 43], particularly those that specify open-set representation [3, 45]. However, OSR methods assume the distinct and fixed class pools for the closed and open sets, which is not feasible in FSOSR, where both sets vary depending on a task. Then, it is hard to straightforwardly apply the conventional OSR methods to FSOSR.

**FSOSR.** Compared to few-shot classification and OSR, FSOSR has been less explored. PEELER [24] introduced FSOSR task and aimed to maximize the entropy of open-set examples with Gaussian embeddings. They employed OSR approach [1] of detecting open-set examples based on the largest class probability. SnaTCHer [16] used transformation consistency that similar examples remain closer after a set-to-set transformation. TANE [15] and D-ProtoNet [19] proposed a task-dependent open-set generator using support examples. Considering low-level features as well as semantic-level ones, GEL [41] designed a local-global energy score to measure the deviation of the input from the closed-set examples. The aforementioned methods primarily employ the representative metric learning approach, ProtoNets [37], and face difficulties in achieving generalization across diverse benchmarks. Our proposed approach, however, involves a two-stage learning process. The first stage focuses on meta-learning to train a task-independent open-set classifier that serves as an effective initialization for the second stage of transfer learning.

# 3 Proposed Method

We present the proposed two-stage transfer learning process, dubbed OAL-OFL. The overall framework of OAL-OFL is depicted in Fig. 2. First, we describe the first stage, open-set aware meta-learning, where a universal FSOSR metric space is obtained by learning the feature encoder and a learnable open-set prototype. Then, we elaborate on the open-set free transfer learning stage, especially suggesting two approaches to resolve the challenges of the absence of open-set examples.

**Problem definition.** The goal of FSOSR is two-fold: to classify an input query into one of the closed-set categories or to reject it as belonging to the associated open-set categories. Formally, an $N$-way $K$-shot FSOSR task can be represented as $\mathcal{T} = \{S, Q, \tilde{Q} \,|\, \mathcal{C}, \tilde{\mathcal{C}}\}$, where $C$ denotes a set of $N$ closed-set categories. The closed set is described by a support set $S$ which includes $K$ examples for each category, i.e. $\{x_i, y_i\}_1^{|S|}$ and $|S| = NK$. Here, $x$ represents the input data, while $y$ indicates its corresponding label. The query set for $C$ is denoted by $Q$. Distinctively, FSOSR also accommodates open-set queries $\tilde{Q}$, which belong to categories, $\tilde{\mathcal{C}}$, that do not intersect with $C$ and are not explicitly defined.

## 3.1 Stage-1: Open-set Aware Meta-Learning

We commence by meta-learning a metric space, which serves as a general starting point to tune the model to a specific target task in the open-set free transfer learning of Stage-2. As in Fig. 2 (a), this meta-learning stage involves the simultaneous training of the feature encoder $f_\theta$ and a learnable open-set prototype $c_\phi$. They are learned using FSOSR tasks drawn from the base training dataset $\mathcal{D}_{\mathrm{bs}}$. Though the definition of open set varies depending on $\mathcal{C}$, the $c_\phi$ is desired to be task-independent across $\forall \mathcal{C}$. Given the complexities involved in learning the task-specific open set classifier during Stage-2, the effective training of a reliable $c_\phi$ serves as a crucial link to bridge the two stages.

In specific, given an $N$-way $K$-shot FSOSR task, $\mathcal{T} \sim \mathcal{D}_{\mathrm{bs}}$, we conceptualize this as the $(N+1)$-way classification problem. In this framework, the $(N+1)$th class is designated as the open-set class. For $n$-th class of $\mathcal{C}$, the classifier is formulated as the prototype $c_n$ which is defined as the mean feature vector over its $K$ examples, i.e., $c_n = \frac{1}{K} \sum_{m=1}^{K} f_\theta(x_m)$ [37]. Here, $f_\theta(x)$ produces a $D$-dimensional feature vector. Concurrently, for the open-set class, the corresponding classifier is characterized by the learnable open-set prototype $c_\phi$.

**(a) Stage 1: Open-Set Aware Meta-Learning (OAL)** 🔥: Learned

Training Task (*N*-way *K*-shot)

**(b) Stage 2: Open-set Free Transfer Learning (OFL)** 🔥: Learned

Target Task (*N*-way *K*-shot)

Figure 2: **Overall training framework of OAL-OFL.** (a) In Stage 1, the feature encoder and a learnable open-set prototype undergo distance-based meta-learning [37] with an additional class representing the open set. (b) In Stage 2, feature encoder and prototypes are further transfer-learned to the target task under an open-set-free condition. Open-set training examples can be alternatively drawn from the base training dataset (green) or from a subset of the closed-set categories that is randomly selected as a pseudo open set (purple).

Subsequently, the classification probability is

$$
\begin{aligned}
p(y = n|x) &= \frac{e^{-a \cdot d(c_\phi, f_\theta(x)) + b}}{e^{-a \cdot d(c_\phi, f_\theta(x)) + b} + \sum_{n'=1}^{N} e^{-d(c_{n'}, f_\theta(x))}} && \text{if } n = N + 1, \\
p(y = n|x) &= \frac{e^{-d(c_n, f_\theta(x))}}{e^{-a \cdot d(c_\phi, f_\theta(x)) + b} + \sum_{n'=1}^{N} e^{-d(c_{n'}, f_\theta(x))}} && \text{Otherwise,}
\end{aligned}
\tag{1}
$$

where $d$ is a distance metric. Specifically, we use the squared Euclidean distance, $d(v, v') = \|v - v'\|^2$. In addition, scalars $a$ and $b$ are also learned to calibrate the distance to $c_\phi$, inspired by D-ProtoNets [19]. This calibration enables that $c_\phi$ is solely accountable for representing open sets of various tasks. For the loss function, we utilize the cross-entropy (CE) loss which is formulated as:

$$
\mathcal{L}_{\text{CE}}(x_i, y_i) = -\log p_\theta(y = y_i|x_i).
\tag{2}
$$

We further utilize the masked CE loss [45], defined as:

$$
\mathcal{L}_{\text{Mask}}(x_i, y_i) = -\log p_{\theta \setminus y_i}(y = N + 1|x_i),
\tag{3}
$$

where $p_{\theta \backslash y_i}$ denotes the $N$-way classification probability, excluding the ground-truth label $y_i$. This loss term effectively creates pseudo open-set tasks by disregarding the true label. More precisely,

$$p_{\theta \backslash y_i}(y = N + 1 | x_i) = \frac{e^{-d_{N+1}}}{e^{-d_{N+1}} + \sum_{n'=1, n' \neq y_i}^{N} e^{-d(c_{n'}, f_\theta(x_i))}},$$

where we let $d_{N+1} = a \cdot d(c_\phi, f_\theta(x_i)) + b$ for brevity.

Overall, we learn $f_\theta$, $c_\phi$, $a$, and $b$ as

$$\theta_1^*, \phi_1^*, a_1^*, b_1^* = \underset{\theta, \phi, a, b}{\arg\min} \{ \mathbb{E}_{x \in \tilde{Q}} \mathcal{L}_{\text{CE}}(x, N+1) + \mathbb{E}_{(x,y) \in Q} \{ \mathcal{L}_{\text{CE}}(x, y) + \mathcal{L}_{\text{Mask}}(x, y) \} \}. \qquad (4)$$

## 3.2 Stage-2: Open-set Free Transfer Learning

In this stage, we consider a target task, denoted as $\mathcal{T}_{\text{te}} = \{ S_{\text{te}}, Q_{\text{te}}, \tilde{Q}_{\text{te}} | \mathcal{C}_{\text{te}}, \tilde{\mathcal{C}}_{\text{te}} \}$, which is also configured as an $N$-way $K$-shot task. Note that $Q_{\text{te}}$ and $\tilde{Q}_{\text{te}}$ are utilized solely for the purposes of inference during test. The initial values of the learnable parameters are inherited from results of Stage-1. Specifically, $f_\theta$, $a$ and $b$ are imprinted as $f_{\theta_1^*}$, $a_1^*$ and $b_1^*$, respectively. We aim to train them alongside $(N + 1)$-way classifiers. These classifiers are represented as $\{ w_1, \cdots, w_{N+1} \}$, where each $w_n \in \mathbb{R}^D$. The first $N$ classifiers are set as the prototypes of the corresponding classes in $C_{\text{te}}$. The open-set classifier $w_{N+1}$ is initialized using $c_{\phi_1^*}$.

To ensure consistency with the meta-learned metric space, we compute the classification probability of each query based on the Euclidean distance as in Eq. (1). Then, during the transfer learning, we optimize $f_\theta$ and the classifier $g_\psi = \{ w_1, \cdots, w_{N+1}, a, b \}$.

Since $\mathcal{T}_{\text{te}}$ lacks cues to learn the open-set classifier $w_{N+1}$, we suggest two approaches: i) sampling from the base training dataset $\mathcal{D}_{\text{bs}}$ of Stage-1, and ii) sampling from the closed set $C_{\text{te}}$ of $\mathcal{T}_{\text{te}}$ itself.

### 3.2.1 Open-set sampling from base training dataset

Notice that in the context of FSL, the closed-set categories $C_{\text{te}}$ in $\mathcal{T}_{\text{te}}$ are exclusive with $\mathcal{D}_{\text{bs}}$. As a result, we can exploit $D_{\text{bs}}$ as a pool of open-set examples for $\mathcal{T}_{\text{te}}$. Specifically, as in the green-colored of Fig. 2(b), we randomly select $M$ examples from $\mathcal{D}_{\text{bs}}$ at every iteration of the transfer learning to serve as open-set examples.

Then, the model is optimized as:

$$\theta_2^*, \psi_2^* = \underset{\theta, \psi}{\arg\min} \{ \mathbb{E}_{(x,y) \in S_{\text{te}}} \mathcal{L}_{\text{CE}}(x, y) + \mathbb{E}_{x \sim D_{\text{bs}}} \mathcal{L}_{\text{CE}}(x, N+1) \}. \qquad (5)$$

We call this unified process of Stage-1 and 2 with the base training dataset as *OAL-OFL*. In the following section, we will introduce the OAL-OFL with a more practical open-set sampling, dubbed *OAL-OFL-Lite*.

### 3.2.2 Pseudo open-set sampling from closed set

In real-world scenarios, $\mathcal{T}_{\text{te}}$ encompasses various categories, and we cannot guarantee that the closed set $C_{\text{te}}$ is not overlapped with the categories of $\mathcal{D}_{\text{bs}}$. Additionally, after the meta-learning stage, the large-scale $\mathcal{D}_{\text{bs}}$ may not be affordable due to practical constraints. To address these issues, we introduce OAL-OFL-Lite which operates with no necessity of $\mathcal{D}_{\text{bs}}$.

Our strategy is the episodic random class sampling from the closed set $C_{\text{te}}$ itself to learn the open set. As exemplified in the purple-colored of Fig. 2(b), we iteratively partition $C_{\text{te}}$ into the mutually exclusive subsets $\acute{C}_{\text{te}}$ and $\tilde{C}_{\text{te}}$. Subsequently, their corresponding support sets $\acute{S}_{\text{te}}$ and $\tilde{S}_{\text{te}}$ extracted from $S_{\text{te}}$ are used to transfer-learn the closed and open sets, respectively. Hence, we call $\tilde{C}_{\text{te}}$ pseudo open set. Through this iterative pseudo open-set sampling, we can effectively learn the open-set classifier as well as the closed-set ones. Then, the model is optimized by the CE losses as

$$\theta_2^*, \psi_2^* = \underset{\theta, \psi}{\arg\min} \{ \mathbb{E}_{(x,y) \in \acute{S}_{\text{te}}} \mathcal{L}_{\text{CE}}(x, y) + \mathbb{E}_{x \in \tilde{S}_{\text{te}}} \mathcal{L}_{\text{CE}}(x, |\acute{C}_{\text{te}}| + 1) \}. \qquad (6)$$

where the open set is mapped to $(|\acute{C}_{\text{te}}| + 1)$th class and detected by $w_{N+1}$ in every iteration. Moreover, we empirically found that the open-set representation tends to be overfitted to a testing task in OAL-OFL-Lite. Hence, we freeze $w_{N+1}$ once it is initialized by the $c_{\phi_1^*}$. For more comprehensive understanding, we include Algorithm 1 in Appendix.

Table 1: **Comparative results on the *mini*ImageNet and *tiered*ImageNet.** Averages of Acc (%) and AUROC (%) over 600 tasks are reported with 95% confidence interval (*: reproduced, †: from [16]). For each case, bold and underlined scores indicate the best and second-best results.

| Methods | *mini*ImageNet | | | | *tiered*ImageNet | | | |
| --- | --- | --- | --- | --- | --- | --- | --- | --- |
| | 1-shot | | 5-shot | | 1-shot | | 5-shot | |
| | Acc. | AUROC | Acc. | AUROC | Acc. | AUROC | Acc. | AUROC |
| ProtoNets[37]† | 64.01 ±0.9 | 51.81 ±0.9 | 80.09 ±0.6 | 60.39 ±0.9 | 68.26 ±1.0 | 60.73 ±0.8 | 83.40 ±0.7 | 64.96 ±0.8 |
| FEAT [42]† | 67.02 ±0.9 | 57.01 ±0.8 | 82.02 ±0.5 | 63.18 ±0.8 | 70.52 ±1.0 | 63.54 ±0.7 | 84.74 ±0.7 | 70.74 ±0.8 |
| PEELER [24]† | 65.86 ±0.9 | 60.57 ±0.8 | 80.61 ±0.6 | 67.35 ±0.8 | 69.51 ±0.9 | 65.20 ±0.8 | 84.10 ±0.7 | 73.27 ±0.7 |
| SnaTCHer-F [16] | 67.02 ±0.9 | 68.27 ±1.0 | 82.02 ±0.5 | 77.42 ±0.7 | 70.52 ±1.0 | 74.28 ±0.8 | 84.74 ±0.7 | 82.02 ±0.6 |
| SnaTCHer-T [16] | 66.60 ±0.8 | 70.17 ±0.9 | 81.77 ±0.5 | 76.66 ±0.8 | 70.45 ±1.0 | 74.84 ±0.8 | 84.42 ±0.7 | 82.03 ±0.7 |
| SnaTCHer-L [16] | 67.60 ±0.8 | 69.40 ±0.9 | 82.36 ±0.6 | 76.15 ±0.8 | 70.85 ±1.0 | 74.95 ±0.8 | 85.23 ±0.6 | 80.81 ±0.7 |
| ATT [15] | 67.64 ±0.8 | 71.35 ±0.7 | 82.31 ±0.5 | 79.85 ±0.6 | 69.34 ±1.0 | 72.74 ±0.8 | 83.82 ±0.6 | 78.66 ±0.7 |
| ATT-G [15] | 68.11 ±0.8 | 72.41 ±0.7 | 83.12 ±0.5 | 79.85 ±0.6 | 70.58 ±0.9 | 73.43 ±0.8 | 85.38 ±0.6 | 81.64 ±0.6 |
| GEL [41] | 68.26 ±0.9 | 73.70 ±0.8 | 83.05 ±0.6 | 82.29 ±0.6 | 70.50 ±0.9 | 75.86 ±0.8 | 84.60 ±0.7 | 81.95 ±0.7 |
| **OAL-OFL** | **69.78** ±0.8 | **73.88** ±0.7 | 85.49 ±0.7 | **83.13** ±0.6 | **71.73** ±0.5 | **75.88** ±0.6 | **86.75** ±0.6 | **83.36** ±0.6 |
| **OAL-OFL-Lite** | 69.15 ±0.8 | 72.21 ±0.9 | **85.61** ±0.6 | 81.11 ±0.6 | 70.80 ±0.9 | 73.67 ±0.7 | 86.66 ±0.6 | 82.22 ±0.6 |

## 4 Experimental Results

### 4.1 Implementation Details

In line with established FSOSR approaches [24, 16, 15, 41], we conducted experiments using ResNet12 [21] as the feature encoder on *mini*ImageNet [40] and *tiered*ImageNet [33] benchmarks, with pre-training of the feature encoder accomplished on the corresponding base training dataset for each benchmark through the well-known FSL method, FEAT [42]. *Mini*ImageNet is comprised of 100 categories each with 600 examples. These categories are divided into 64, 16, and 20 classes for training, validation, and testing purposes, respectively [32]. *Tierd*ImageNet consists of 608 categories and a total of 779,165 samples, which are divided into 351, 97, and 160 classes for training, validation, and testing, respectively. More training details are in Appendix.

**Evaluation protocol.** We use the predicted probability for $(N + 1)$th class as an open-set score and report threshold-free measurement AUROC [5, 8]. For closed-set evaluation, we predict the class with the highest probability among closed-set categories and report top-1 accuracy (Acc).

**Compared methods.** We conduct a comparative analysis of our OAL-OFL method against various existing approaches, encompassing both FSL methods such as [37, 42] and FSOSR methods [24, 16, 15, 41].

### 4.2 Comparative Assessment

Table 1 presents a comparison between our OAL-OFL and baselines in 5-way {1, 5}-shot settings on *mini*ImageNet and *tiered*ImageNet. FSL approaches [37, 42] were not designed for FSOSR, we applied them to FSOSR in a straightforward manner as in [16]. In brief, in the FSL methods, the open-set detection score is computed by taking the negative of the largest classification probability [23]. Compared to the FSOSR methods, PEELER, SnaTCHers and ATT, both OAL-OFL and OAL-OFL-Lite show better generalized capabilities, as reflected by their Acc and AUROC across datasets.

Nevertheless, notice that there has been no further performance improvement since GEL, highlighting the challenges of progress in FSOSR. This is largely due to the inherent trade-off between Acc and AUROC—two metrics that often conflict. For instance, although GEL achieves the best AUROC among prior methods, it falls short of ATT-G in terms of Acc across both 1 and 5-shot settings. More specifically, it is observed that the previous approaches exhibit a particular bias towards miniImageNet. ATT-G improves ATT by using the base class prototypes to enhance closed-set prototypes. Nevertheless, it is even worse than our OAL-OFL-Lite which does not use the base training dataset. GEL makes predictions for both pixel-wise and semantic-wise by adding a 2D convolutional block on top of the SnaTCHer-F. It attained the previous SOTA in *mini*ImageNet, but not in all measures. Whereas we achieve SOTA performance *in all cases*. In 1-shot setting, we slightly outperform GEL in AUROC, but are much better in Acc (1.52% in *mini*ImageNet and 1.23% in *tiered*ImageNet). Notice that in FSOSR, both closed-set classification and open-set detection are important. In 5-shot setting, our method shows clear SOTA performance in both Acc and AUROC. In specific, 2.59% Acc

Table 2: **Ablation analysis on stages in the proposed OAL-OFL**.

| Stage-1 | Stage-2 | 1-shot | | 5-shot | |
|---|---|---|---|---|---|
| | | Acc. | AUROC | Acc. | AUROC |
| Naive TL to closed set | | 66.05 | 62.38 | 84.21 | 69.07 |
| Stage-2 only + regul. | | 56.97 | 67.78 | 78.98 | 76.72 |
| | ✓ | 56.72 | 67.47 | 78.59 | 76.64 |
| ✓ | | 67.95 | 71.81 | 82.76 | 80.91 |
| ✓ | ✓ | 69.78 | 73.88 | 85.49 | 83.13 |

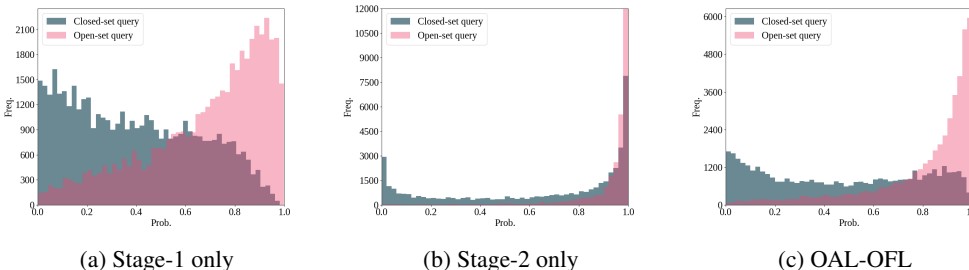

| (a) Stage-1 only | (b) Stage-2 only | (c) OAL-OFL |
|---|---|---|

Figure 3: **Distribution of the classification probabilities of closed-set and open-set queries to the open-set class.** Closed-set and open-set queries are represented in green and red, respectively.

and 0.84% AUROC in *mini*ImageNet, and 2.15% Acc and 1.41% in *tiered*ImageNet. Namely, when only a few examples per class are increased (one to five), the proposed open-set free transfer learning more faithfully makes it beneficial. Also, even without the base training dataset, OAL-OFL-Lite surpasses or is comparable to the existing FSOSR methods in both datasets. Therefore, we conclude that our two-stage approach successfully pioneers transfer learning for FSOSR. In addition, cross-domain comparative results are provided in Appendix.

## 4.3 Analysis

We extensively analyze the proposed method on *mini*ImageNet. More analyses are in the Appendix.

**Difficulty of TL in FSOSR.** To see the efficacy of the respective stage of our method especially in FSOSR, we first analyze the proposed OAL-OFL comparing with two straightforward transfer learning schemes in Table 2. First, in 'Naive TL to closed set' of the first row, we conduct a naive transfer learning [34] without the base training dataset. Namely, the model is fit to $N$-way closed-set classification through linear regression [4], and rejects the open-set queries thresholding max-probability [1]. This naive approach shows severely low performance, which indicates the difficulty of FSOSR transfer learning. Second, in 'Stage-2 only + regul.,' the pseudo-label-based regularization exploits the base training dataset to compute the Kullback–Leibler divergence between the pre-trained model and transfer-learned model, which showed promising results in transfer learning for FSL [17]. Skipping Stage-1, we can apply this regularization technique during transfer learning which is 'Stage-2 only + regul.' This approach (single-stage + regularization) shows only slight improvement compared to the case of only Stage-2 (1st row of the lower part of Table 2). Whereas, our OAL-OFL achieves meaningful transfer learning results by combining the two stages.

**Two-stage training (OAL-OFL).** To demonstrate the impact of the proposed two-stage training approach, we first ablate the stages in Table 2 for OAL-OFL. When Stage-1 is skipped, the open-set classifier $w_{N+1}$, and scalars $(a, b)$ are initialized in random and $(1,0)$, respectively in Stage-2. Relying solely on Stage-2 causes overfitting to the training examples. Then, it leads to highly degenerated results of 56.72% and 78.59% for 1- and 5-shot closed-set classification Acc, respectively, and 67.47% and 76.64% for 1- and 5-shot open-set detection AUROC, respectively. These results highlight that the open-set aware meta-learning of Stage-1 is critical for the success of transfer learning in FSOSR.

Notice that, in ATT-G [15], the base training dataset calibrates the closed-set classifiers even in testing phase, but it marginally improves ATT [15] (see Table 1). Whereas, in the proposed OAL-OFL, the use of the base training dataset during the open-set free training of Stage-2 gives notable

Table 3: **Ablation analysis for our OAL-OFL-Lite in Stage-2.**

| | 1-shot | | 5-shot | | | | Pseudo open set | Freeze $w_{N+1}$ | 1-shot | | 5-shot | |
|---|---|---|---|---|---|---|---|---|---|---|---|---|
| | Acc. | AUROC | Acc. | AUROC | | | | | Acc. | AUROC | Acc. | AUROC |
| Naive TL to closed set | 66.05 | 62.38 | 84.21 | 69.07 | | OAL-OFL -Lite | | | 68.14 | 71.52 | 84.36 | 80.88 |
| Stage-2 only | 57.72 | 68.13 | 78.43 | 79.36 | | | ✓ | | 68.20 | 71.33 | 85.12 | 80.61 |
| OAL-OFL-Lite | 69.15 | 72.21 | 85.61 | 81.11 | | | ✓ | ✓ | 69.15 | 72.21 | 85.61 | 81.11 |

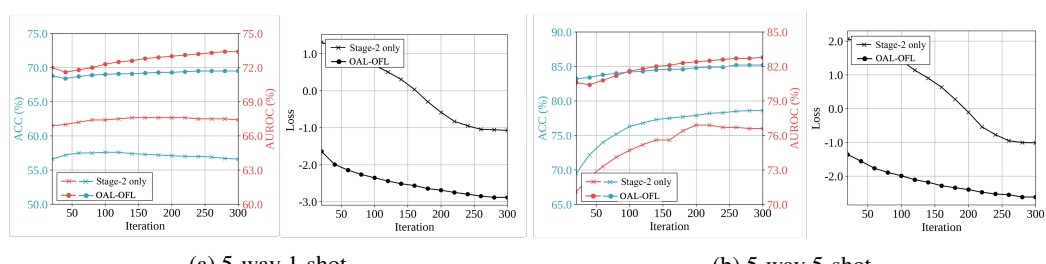

(a) 5-way 1-shot  (b) 5-way 5-shot

Figure 4: **Plots of OAL-OFL on Acc (%), AUROC (%), and loss (logarithmic scale) on iterations in Stage-2** (Best viewed in color).

improvement. Applied on top of Stage-1, Stage-2 attains notably improved results by 1.83% and 2.70% in Acc, and 2.07% and 2.22% in AUROC for 1- and 5-shot respectively.

In Fig. 3, we present a visualization of the distributions of classification probability towards the open-set class, derived from both closed-set and open-set queries. The visualization reveals that, in the scenario labeled as 'Stage-2 only,' where the open-set aware meta-learning is absent, a majority of the queries are assigned high probabilities of belonging to the open-set, irrespective of their actual membership to the closed or open set. Converserly, in 'Stage-1 only' scenario, a significant number of closed-set queries are still attributed probabilities exceeding 0.5. On the other hand, in OAL-OFL, it is observed that the majority of open-set queries attain significantly high probabilities of being classified as the open-set, while the probabilities for closed-set queries are effectively suppressed. This distinct separation in the classification probabilities for open and closed-set queries allows us to understand and verify the robustness of OAL-OFL in dichotomizing between open and closed sets.

**Two-stage training (OAL-OFL-Lite).** The left part of Table 3 shows the stage ablation for OAL-OFL-Lite. On top of the same Stage-1 model of OAL-OFL, the open-set free transfer learning of OAL-OFL-Lite also achieves a notable improvement on the Stage-1 model in Table 2. As shown in the ablation study on OAL-OFL, 'Stage-2 only' results in a large performance drop compared to the OAL-OFL-Lite of the third row. In the following sections, we study the key components of the open-set free transfer learning of the proposed OAL-OFL and OAL-OFL-Lite.

**Pseudo open set & Frozen $w_{N+1}$.** OAL-OFL-Lite deals with the challenge of the absence of the base training dataset by the episodic sampling of the pseudo open set and freezing $w_{N+1}$. The right part of Table 3 shows the importance of the two components. Without both (the first row), the model focuses on learning closed-set classification, showing better Acc and lower AUROC than using only Stage-1 in Table 2. Although the pseudo open-set sampling is used, transfer learning the open-set classifier is still challenging as demonstrated in the second row. Hence, as in the last row, we freeze the open-set classifier and perform episodic pseudo open-set sampling.

**Stage-1 to prevent overfitting.** Fig. 4 visualizes the changes of Acc, AUROC, and loss values during Stage-2 training. To see the impact of Stage-1 on Stage-2, we plot 'Stage-2 only' and our complete OAL-OFL. Fig. 4(a) right shows the loss curves. OAL-OFL starts from a pretty lower loss than 'Stage-2 only.' And both 'Stage-2 only' and OAL-OFL converge, but 'Stage-2 only' is saturated at a higher loss. Also, OAL-OFL shows better performance over all the iterations in the testing ACC and AUROC of Fig. 4(a) left. We can see a similar trend in 5-shot. From this result, we can infer that our Stage-1 can provide a favorable starting point, while mitigating the overfitting problem of FSOSR transfer learning.

**Analysis on Stage-1.** We exploited the scalar factors, $a$ and $b$, and masked loss $L_{mask}$ to facilitate learning task-independent open-set prototype in the meta-learning of Stage-1. Table 4 ablates these

Table 4: **Ablation analysis for Stage-1.**

| $\mathcal{L}_{\text{mask}}^1$ | $(a,b)$ | 1-shot | | 5-shot | |
|---|---|---|---|---|---|
| | | Acc. | AUROC | Acc. | AUROC |
| | | 66.07 | 70.72 | 82.01 | 80.14 |
| ✓ | | 66.45 | 71.06 | 81.75 | 80.31 |
| ✓ | ✓ | 67.95 | 71.81 | 82.76 | 80.91 |

Table 5: **Impact of the base training dataset varying the number of classes in Stage-2.**

| #Base categories | 1-shot | | 5-shot | |
|---|---|---|---|---|
| | Acc. | AUROC | Acc. | AUROC |
| 0 (OAL-OFL-Lite) | 69.15 | 72.71 | 85.61 | 81.11 |
| 1 | 66.41 | 69.62 | 83.66 | 79.62 |
| 6 | 66.72 | 69.71 | 83.15 | 78.34 |
| 32 | 68.30 | 71.84 | 84.51 | 81.34 |
| 64 (OAL-OFL) | 69.78 | 73.88 | 85.49 | 83.13 |

Table 6: **Effect of meta-learning on the classifier of Stage-2**. (Rand: random, Meta: Stage-1 meta-learned).

| Classifier Initialization | | OAL-OFL-Lite | | | | OAL-OFL | | | |
|---|---|---|---|---|---|---|---|---|---|
| | | 1-shot | | 5-shot | | 1-shot | | 5-shot | |
| $\{w_n\}_{n=1}^N$ | $w_{N+1}$ | Acc. | AUROC | Acc. | AUROC | Acc. | AUROC | Acc. | AUROC |
| Rand | Rand | 40.34 | 35.29 | 50.90 | 27.19 | 63.03 | 67.52 | 79.97 | 75.95 |
| Meta | Rand | 57.72 | 68.13 | 78.43 | 79.36 | 69.02 | 70.87 | 85.32 | 78.50 |
| Meta | Meta | 69.15 | 72.71 | 85.61 | 81.11 | 69.78 | 73.88 | 85.49 | 83.13 |

components. All the components are useful, and the degraded performance in the ablated ones means the difficulty of meta-learning the task-independent open-set prototype.

**Amount of $\mathcal{D}_{\text{bs}}$ in Stage-2.** In Table 5, we identify the impact of the variety of categories in $\mathcal{D}_{\text{bs}}$ on the quality of the OFL. By reducing the number of categories in $\mathcal{D}_{\text{bs}}$, the overall results are degraded. Rather, when the number of the base categories is lower than the half, OAL-OFL-Lite shows better results. Hence, it is crucial to configure the open-set training dataset from diverse base categories.

**Meta-learned classifier.** In OAL-OFL and OAL-OFL-Lite, we utilize the meta-learned feature encoder and open-set prototype to initialize the closed-set and open-set classifiers. As in Table 6, although the feature encoder starts from the meta-learned weights, the transfer-learned model suffers from severe performance degradation due to random initialization of both open and closed-set classifiers (Rand-Rand in Table 6). In OAL-OFL, when the closed-set classifiers are initialized by the class-wise prototypes from the meta-learned encoder (Meta-Rand), the closed-set classification Acc is recovered, but open-set AUROC is not. In OAL-OFL-Lite, both Acc and AUROC are quite lower than the case of complete meta-learned weight initialization. Further, when both closed and open-set classifiers are not initialized by the meta-learned weight, the open-set recognition ability of OAL-OFL is worse than OAL-OFL-Lite where classifiers are initialized by the meta-learned weights. Hence, the open-set aware meta-learning is crucial for the both closed- and open-set classifiers.

**Cost for Stage-2.** As mentioned in Sec. 4.1, we train the model with 20,000 tasks, an iteration per task in Stage-1. This is *de facto* standard in FSOSR methods. In our Stage-2 of both OAL-OFL and OAL-OFL-Lite, the model is further learned during 300 iterations. It is only 1.5% of Stage-1. While the model is not more generalized in Stage-1 with the 1.5% additional iterations, our Stage-2 yields notable improvement as in Table 2 through a tiny extra training cost.

## 5    Conclusions

This work introduced the two-stage learning approach for FSOSR called *open-set aware meta-learning (OAL) on a base dataset and open-set free transfer learning (OFL)* on testing tasks. Our findings highlight the significance of the meta-learned FSOSR metric space, which serves as a general starting point in transfer learning and enables us to take advantage of transfer learning. For the open-set free transfer learning, we provided two suggestions to configure open-set training data: 1) sampling from the base dataset, and 2) sampling from the testing task itself. The latter one considers a more practical scenario, not relying on the base dataset during inference. Both are beneficial for generalizing the model to an FSOSR testing task. As a result, we achieved SOTA performance on two benchmark datasets, namely *mini*ImageNet and *tiered*ImageNet.

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

## A  Algorithm

---

**Algorithm 1** OAL-OFL

---

**Stage-1.** Meta-Learn FSOSR Metric Space
**Require:** Dataset $\mathcal{D}_{\text{bs}}$, encoder $f_\theta$.

1: Initialize $c_\phi$, $a$, and $b$.
2: **for** # of training episodes **do**
3:     Sample a task $\mathcal{T} = \{S, Q, \tilde{Q} | \mathcal{C}, \tilde{\mathcal{C}}\}$             ▷ A training episode
4:     $c_n = \frac{1}{K} \sum_{m=1}^{K} f_\theta(x_m)$, for $n \in \mathcal{C}$ and $y_m = n$      ▷ Prototype
5:     $c_1, \cdots, c_N$ = LayerTaskNorm$(c_1, \cdots, c_N)$     ▷ Apply LayerTaskNorm [16]
6:     Update $\theta, \phi, a, b$ =$\arg\min_{\theta,\phi,a,b}\{\mathbb{E}_{x\in\tilde{Q}}\mathcal{L}_{\text{CE}}(x, N + 1) + \mathbb{E}_{(x,y)\in Q}\{\mathcal{L}_{\text{CE}}(x, y) +$
   $\mathcal{L}_{\text{Mask}}(x,y)\}\}$
7: **end for**
**Ensure:** $\theta_1^*, \phi_1^*, a_1^*, b_1^* = \theta, \phi, a, b$

**Stage-2.** Transfer Learning
**Require**: Dataset $\mathcal{D}_{\text{bs}}$ and $\mathcal{D}_{\text{te}}$, $\theta_1^*, \phi_1^*, a_1^*, b_1^*$

1: Initialize: $f_\theta$ with $\theta_1^*$
2: $c_n = \frac{1}{K} \sum_{m=1}^{K} f_\theta(x_k)$, for $n \in \mathcal{C}_{\text{te}}$ and $y_m = n$
3: Initialize: $g_\psi$ with $\phi_1^*, a_1^*, b_1^*, c_1, \cdots, c_N$.
4: **for** # of iterations **do**
5:     Update $\theta, \psi = \arg\min_{\theta,\psi}\{\mathbb{E}_{(x,y)\in S_{\text{te}}}\mathcal{L}_{\text{CE}}(x, y) + \mathbb{E}_{x\sim D_{\text{bs}}}\mathcal{L}_{\text{CE}}(x, N + 1)\}$
6: **end for**
**Ensure:** $\theta_2^*, \psi_2^* = \theta, \psi$

---

## B  Training details

Table 7: **Training details.**

|  | Stage-1 | Stage-2 |
|---|---|---|
| Optimizer | SGD | SGD |
| Optimizer momentum | 0.9 | 0.9 |
| Weight decay | 5e-4 | 5e-4 |
| Learning rate (LR) | 2e-4 for $f_\theta$, 2e-5 for others | 2e-4 for $f_\theta$, 2e-3 (OAL-OFL) 2e-4 (OAL-OFL-Lite) for others |
| LR decaying | multi. 0.5 every 40 episodes | None |
| # of episodes (iterations) | 20,000 | 300 |

Stage-1 employs episodic learning on the base training dataset, $\mathcal{D}_{\text{bs}}$, and trains the model for 20,000 episodes using the SGD optimizer with the Nesterov momentum [26] and the weight decay of 5e-4. The learning rates are initialized to 2e-4 and 2e-5 for the encoder and the learnable open-set prototype, respectively, and then decayed by multiplying 0.5 every 40 iterations. Each episode consists of a combination of closed-set categories and an equal number of open-set categories, each containing 15 queries. The distance adjusting factors $a$ and $b$ are initialized to 1 and 0, respectively. Layer-task normalization [16] is used as the embedding adaptation, which is discarded after the initialization of the classifiers in Stage-2.

In Stage-2, the proposed transfer learning is performed for 300 iterations on each testing task using the SGD optimizer. For OAL-OFL, the learning rates for the feature encoder and classifier are set to 2e-4 and 2e-3, respectively. For OAL-OFL-Lite, the learning rate is 2e-4 for both. The weight decay and momentum are set to 5e-4 and 0.9. For episode configuration in OAL-OFL-Lite, one closed-set class is randomly sampled as the pseudo open set at every iteration.

**Pretrain encoder.** In alignment with the latest developments in few-shot classification and FSOSR methods [17, 16, 15], we have pre-trained the encoder, denoted as $f_\theta$. This was achieved using a linear classifier that categorizes all classes in the dataset $\mathcal{D}_{\text{bs}}$. The training process spanned 500 epochs utilizing the SGD optimizer. We employed cross-entropy loss complemented by self-knowledge distillation [12], and integrated rotation as per [15]. Additionally, data augmentation was implemented

using the mixup [44]. This training approach aligns our work with the SOTA methods [15, 41] in the field, facilitating fair and relevant comparisons.

Additionally, the specifics of the training setup for both Stage-1 and Stage-2 of our methodologies are detailed in Table 7.

## C   Classifier design in Stage-2.

Table 8: **Linear *vs* Euclidean classifiers in OFL**.

|  | 1-shot | | 5-shot | |
| Classifier | Acc. | AUROC | Acc. | AUROC |
| --- | --- | --- | --- | --- |
| Linear | 68.01 | 72.79 | 84.72 | 81.03 |
| Euclidean (Ours) | 69.78 | 73.88 | 85.49 | 83.13 |

Stage-1 is based on the squared Euclidean distance as the distance metric. In order to maintain consistency in the metric space, we compute the classifier outputs under the same metric in Stage-2 of OAL-OFL. In Table 8, we compared our Euclidean classifier with the linear classifier, meaning the significance of metric-space consistency for the collaborative use of the different two learning approaches.

## D   Analysis on OAL-OFL-Lite varying iterations

Fig. 5 plots the AUROC, ACC, and loss curves for the proposed OAL-OFL-Lite framework compared to its Stage-2 (open-set free transfer learning) only version. These metrics are plotted over training iteration on the *mini*ImageNet dataset. We can identify that the open-set aware meta-learning plays a crucial role in preventing overfitting and enhancing the efficacy of transfer learning in OAL-OFL-Lite as its impact in OAL-OFL.

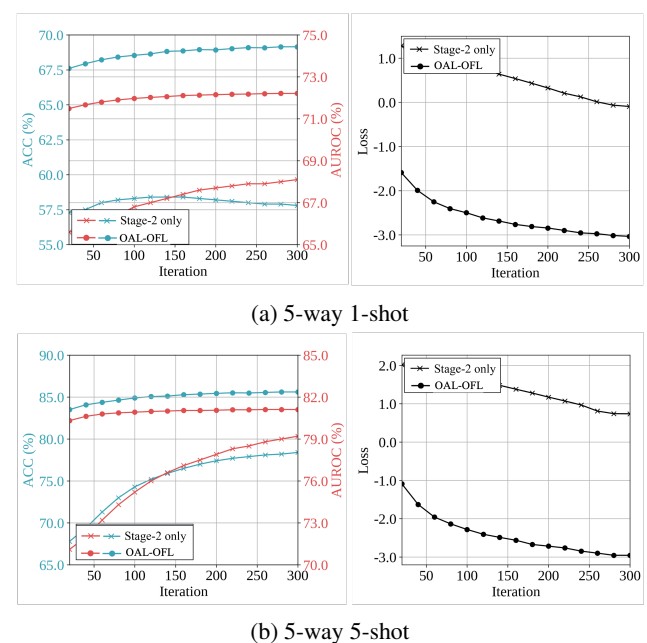

(a) 5-way 1-shot

(b) 5-way 5-shot

Figure 5: Plots of OAL-OFL-Lite on accuracy (%), AUROC (%), and loss (logarithmic scale) increasing iterations in Stage-2 on the 5-way 1- and 5-shot settings of the *tiered*ImageNet.

Table 9: Open-set detection on 600 testing tasks of *tiered*ImageNet 5-way 5-shot setting

| Method | Accuracy (%) | F1-score | AUROC (%) |
|---|---|---|---|
| SnaTCHer-F [16] | 55.44 | 0.6904 | 82.02 |
| SnaTCHer-T [16] | 54.42 | 0.6865 | 82.03 |
| SnaTCHer-L [16] | 70.82 | 0.7376 | 80.81 |
| OAL-OFL | 74.91 | 0.7697 | 83.36 |
| OAL-OFL-Lite | 73.66 | 0.7530 | 82.22 |

## E  Threshold-free open-set detection

To evaluate the performance of open-set recognition, we utilized the widely-used AUROC metric, which is valued for its threshold-free nature. However, the task-specific adjustment of the threshold values for the detection of unseen (open-set) sample poses practical challenge in real-world scenarios.

In our research, the OAL-OFL yields a dedicated open-set classifier. This allows us to exploit the possibility associated with open-set classification for detecting open-set instances. Consequently, a feasible method for setting a threshold involves classifying an input as open-set if its probability of being an open-set class exceeds that of being closed-set classes.

Aligning with our concern, Jeong *et al.* [16] proposed a distinctive approach. They primarily utilized the distance between the average of prototypes and the farthest prototype to differentiate between the closed and open-set data. We conducted a comparative analysis of their results with our OAL-OFL and OAL-OFL-Lite in the context of threshold-free open-set detection in Table 9. The results highlight the effectiveness of our proposed method in detecting open-set instances without the need of any manually selected threshold, thereby underlining its practical utility and effectiveness.

## F  Ablation experiments on *tiered*ImageNet

Table 10: **Ablation analysis on stages in the proposed OAL-OFL on the *tiered*ImageNet**. Accuracy (%) and AUROC (%) are reported on the 5-way 1- and 5-shot settings.

| Stage-1 | Stage-2 | 1-shot | | 5-shot | |
|---|---|---|---|---|---|
| | | Acc. | AUROC | Acc. | AUROC |
| | ✓ | 64.73 | 70.98 | 83.68 | 80.21 |
| ✓ | | 70.96 | 75.05 | 84.78 | 82.08 |
| ✓ | ✓ | 71.40 | 75.45 | 86.75 | 83.36 |

Table 11: **Ablation analysis for the proposed OAL-OFL-Lite in Stage-2 on the *tiered*ImageNet.** Accuracy (%) and AUROC (%) are reported on the 5-way 1- and 5-shot settings.

| | | 1-shot | | 5-shot | |
|---|---|---|---|---|---|
| | | Acc. | AUROC | Acc. | AUROC |
| Naive TL to closed set | | 71.36 | 58.67 | 86.78 | 61.85 |
| Stage-2 only | | 65.07 | 68.70 | 85.03 | 77.67 |
| OAL-OFL-Lite | | 70.80 | 73.67 | 86.66 | 82.22 |
| Pseudo open-set | Freeze $w_{N+1}$ | | | | |
| | | 70.59 | 72.84 | 85.51 | 80.33 |
| ✓ | | 70.51 | 73.24 | 86.45 | 81.92 |
| ✓ | ✓ | 70.80 | 73.67 | 86.66 | 82.22 |

We conducted ablation studies for both OAL-OFL and OAL-OFL-Lite on *tiered*ImageNet dataset. The results are presented in Tables 10 and 11, respectively.

## G   Cross-domain evaluation

Table 12: **Cross-domain assessment.** Accuracy (%) and AUROC (%) are reported.

| | tierd-CUB | | CUB-tierd | | CUB-CUB | |
|---|---|---|---|---|---|---|
| | 1-shot | 5-shot | 1-shot | 5-shot | 1-shot | 5-shot |
| PEELER | 69.5, 67.6 | 84.1, 76.1 | 55.9, 53.8 | 75.0, 64.6 | 59.4, 58.6 | 78.4, 66.0 |
| SnaTCher | 70.8, 83.7 | 85.2, 90.2 | 59.4, 98.6 | 79.0, 99.5 | 59.7, 64.8 | 78.7, 70.7 |
| Ours | 71.3, 88.8 | 87.2, 92.4 | 60.8, 99.0 | 82.1, 99.8 | 61.9, 67.2 | 81.1, 73.0 |

We provide the cross-domain results following SnaTCher. The meta-trained model with tieredImageNet is transfer-learned on three CUB200 cross-domain scenarios. Open-closed configuration are as follows: tieredImageNet-CUB200, CUB200-tieredImageNet, CUB200-CUB200. In this experiment, at least one of the closed or open sets is in an unseen domain during the Stage-1, and each task is 5-way 5-shot. Compared to the baselines whose cross-domain results are reported, we attain consistently better results (Acc, AUROC) in all the configurations.

## H   Discussions

The results achieved with OAL-OFL and OAL-OFL-Lite demonstrate that transfer learning holds great potential for FSOSR. The superior performance of OAL-OFL over OAL-OFL-Lite can be attributed to open-set sampling from the base training dataset. As we consider future research directions, several discussion points come to the forefront:

(i) The efficacy of OAL-OFL can be negatively impacted with a scarcity of open-set data. Although OAL-OFL-Lite provides an effective solution in situations where the base training dataset is unavailable, it maintains the open-set classifier static throughout the transfer learning stage. Thus, there is an opportunity for future research to delve into methods for optimizing the use of pseudo open-set data further refine the open-set classifier.

(ii) In practical scenarios, it is often observed that the support examples gathered in testing environments are of sub-optimal quality. Moving forward, future research endeavors is expected to focus on addressing the complexities associated with FSOSR transfer learning, particularly under conditions where the support examples are disorganized. This includes scenarios where the examples originate from varied data distribution, rather than being derived from a unified, coherent dataset, and may even incorporate incorrectly labeled examples.

