# OpenReview forum: "Unlocking Transfer Learning for Open-World Few-Shot Recognition"
_NeurIPS.cc/2025/Workshop/Reliable_ML — NeurIPS 2025 - Reliable ML Workshop_

### Official Review · Reviewer_AfGh · 2025-09-19
**Two-stage transfer learning (OAL-OFL) improves few-shot open-set recognition by introducing an open-set prototype in meta-learning and open-set-free adaptation.**

**Rating:** 7
**Confidence:** 4

**Review:**

The paper proposes a two-stage framework for few-shot open-set recognition (FSOSR). Stage-1 (OAL) meta-learns a feature encoder and a learnable open-set prototype with calibration, using masked CE to simulate open conditions. Stage-2 (OFL) adapts to target tasks without open labels, either sampling open data from the base set (OAL-OFL) or constructing pseudo-open classes (Lite). Experiments on miniImageNet and tieredImageNet show state-of-the-art accuracy and AUROC, with only ~1.5% additional training.

Strengths:
	•	Well-motivated two-stage design bridging transfer learning and FSOSR.
	•	Nice mechanism: open-set prototype with calibration, masked CE.
	•	Practical Lite variant for base-data restrictions.
	•	Thorough ablations on stages, metric choice, base data size.
	•	Low adaptation cost (300 iterations).

Weaknesses / Limitations
        .  	Code not released; reproducibility relies only on textual descriptions.
	.	Open-set compressed into a single prototype, fragile under multi-modal unknowns.
	.	Pseudo-open ≠ true OOD; robustness overstated.
	.	No thresholding/calibration study (AUROC alone insufficient).
	•	Dependence on base data for best results, weaker Lite variant.
	•	Limited comparison with richer metric/energy-based heads.
	•	No theoretical guarantees despite geometric structure.
	•	Some Stage-2 details missing for reproducibility.

Suggestions for Authors
        1.	Report all Stage-2 hyperparameters and **release code**.
	2.	Test mixture/energy-based open-set heads.
	3.	Add true OOD stress tests (e.g., Places, COCO) with FPR@95%TPR, AUPR.
	4.	Analyze calibration of $(a,b)$ across tasks; include ECE.
	5.	Refine pseudo-open sampling (distance-based splits, augmentations).
	6.	Explore privacy-preserving OFL (feature replay, distillation).

---

### Official Review · Reviewer_4jjX · 2025-09-19
**Interesting method that combines meta-learning and few-shot learning**

**Rating:** 5
**Confidence:** 2

**Review:**

This paper discusses how meta-learning with few-shot learning can be used to improve performance on a FSOSR task. However, the main difficulty for me while reading the paper is that the tasks themselves were not defined adequately for me to appreciate the significance of the approach, and it is be helpful if the authors made the exposition more friendly for readers that are uninitiated with the relevant literature.

Regardless of this, the paper is presented fairly clearly with informational diagrams to describe how OAL and OFL stages are performed. A discussion about the computational cost of this method (since it performs two stages of learning) relative to the baseline would help better position this contribution as well.